# Barriers and opportunities to preventing residential bird–window collisions

**Anastasia J. V. Lysyk**[1‡*], **Aalia I. Khan**[1‡], **Deborah Conners**[2], **Rachel T. Buxton**[1,3]

**1** Department of Biology, Carleton University, Ottawa, Canada, **2** Department of Sociology and Anthropology, Carleton University, Ottawa, Canada, **3** Institute of Environmental and Interdisciplinary Science, Carleton University, Ottawa, Canada

‡ These authors shares first authorship on this work.
* stashalysyk@cmail.carleton.ca

## Abstract

Collisions with windows are a leading source of avian mortality in North America. Window treatment options are commercially available; however, these solutions are rarely used. To investigate knowledge and perceptions of bird–window collisions, willingness to treat windows, and barriers and solutions to treating windows we conducted a survey of residents in Ottawa, Canada. Of 422 survey respondents, 90.7% had previously heard of bird–window collisions, 58.5% had previously observed a collision, 88.0% consider collisions with windows to be an issue in Ottawa, and 87.0% were willing to treat their windows. For all survey respondents, the top barriers reducing willingness to treat windows included the perception that birds infrequently or never collide with windows, aesthetics, and wanting a clear view from windows. For those willing to treat their windows, lack of time was the most identified barrier (38.2%), while for those unwilling to treat their windows, the need for more evidence that bird–window collisions require action was most identified (49.1%). Top potential solutions were provision of free materials, aesthetically pleasing materials, and clear instructions. Our results suggest that Ottawa residents are generally willing to treat their windows at home and we identify key barriers between willingness and implementation. To encourage bird-friendly window treatment at a wider scale, we suggest targeted messaging highlighting the impact of low-rise housing in driving the problem and the solution to bird–window collisions. Our results also highlight the opportunity for advocacy groups to aid residents in overcoming practical barriers to treating their windows.

## Introduction

Birds in North America are experiencing dramatic declines due to a variety of human-driven threats [1]. Each year in North America, billions of birds are killed by colliding with windows, making it one of the largest sources of direct avian mortality [2,3].

**Data availability statement:** All relevant data are within the manuscript and in the Supporting information files. The online repository is available at: https://doi.org/10.17605/OSF.IO/GBHCF.

**Funding:** Funding was provided to RTB, AJVL, and AK through three funding sources. 1) Natural Science and Engineering Research Council of Canada grant (RGPIN 04888) (https://www.nserc-crsng.gc.ca/index_eng.asp). 2) the Kenneth Molson Foundation (https://fondationmolson.org/en/). 3) Environment and Climate Change Canada (GCXE24S042) (https://www.canada.ca/en/environment-climate-change.html). Funders did not play any role in study design, data collection and analysis, decision to publish, or preparation of the manuscript.

**Competing interests:** The authors have declared that no competing interests exist.

Several studies and grassroots programs have quantified the magnitude of the issue of bird–window collisions. Citizen science programs in multiple countries led by members of the public who patrol cities for birds that collide with windows have demonstrated the dramatic numbers of annual bird–window collisions [3]. This includes the Fatal Light Awareness Program (FLAP), that has documented over 100,000 bird–window collisions across the globe [4]. The resulting data have been used in research, successful lawsuits, advocacy programs, and changes to municipal green standards in Canada [5] and beyond [3,6].

Birds collide with windows because they do not perceive them as barriers in the environment due to the transparent and reflective properties of glass [7]. Solutions like window treatments are now commercially available to prevent bird–window collisions, including window films and decals [8–10]. Patterns of uniformly spaced visual markers that cover a window surface are known to provide a barrier that results in avoidance by birds [8–10]. Efficacy of these visual patterns has been found to range from 92–95% reduction in collisions at low-rise buildings (<14 storeys) [9,10] and by 84% at a rural residential building [11].

While bird–window collisions occur at all types of buildings, houses account for approximately 90% of all collisions, as they are relatively more abundant than tall commercial buildings [2]. Some conservative estimates indicate window-related mortality rates at residences are between 1.2 and 2.4 birds annually at Canadian single-and semi-detached houses [12]. Other studies have estimated collision mortalities to be 0.3–15.7 birds per year per residence [2] and reported upwards of 16–43 annual collisions for the worst residences [13]. Although many people have observed bird–window collisions at their homes (39% of respondents in [13]), awareness and uptake of preventative measures remains rare [14].

Despite an understanding of the magnitude of the problem and the effectiveness of solutions, there have been few coordinated efforts to address the bird–window collisions at scale in Canada and elsewhere [14]. Treating windows can be a contentious issue, as glass is valued for its aesthetically appealing transparency that allows landscape views [15]. Retro-fitting windows can be cost-prohibitive and logistically challenging, requiring adhering decals to the outside surface of sometimes difficult-to-reach windows. Although bird mortality resulting from window collisions are unlawful in Canada under the *Migratory Birds Convention Act, 1994* [16], enforcement typically focuses on urban commercial buildings despite residential homes being a key source of bird–window collisions [2,13]. Given the potential for window treatments at residential homes to reduce up to 90% of bird–window collisions, it is important to understand how to motivate residents to implement bird–window protection measures.

In a survey of the Canadian public, one study found that Canadian's willingness to pay to reduce bird–window collisions at their home was positively influenced by respondents' interest in birds, previous donations to conservation organizations, and demographic factors like age and income [12]. In another study exploring perceptions of bird–window collisions in the US, architects, homeowners, and conservation practitioners indicated a general receptivity to management measures [17]. However,

despite awareness and receptivity, most people do not understand the magnitude of the issue nor available solutions or resources [15].

Community science programs to raise awareness about bird–window collisions have proliferated across North America [3], including local awareness-raising campaigns about window treatments [18]. Strategies include increasing knowledge and public information, assuming that the more informed people are, the more likely they are to take action [19]. However, many factors influence the likelihood of people engaging with conservation behaviours, including relationships to nature, social norms, and the perceived ability for one's actions to be impactful [19,20]. To guide efforts to increase uptake of bird-friendly window treatments, we aimed to investigate key barriers identified by residents in Ottawa to treat their windows. Specifically, our objectives were to: 1) explore the current knowledge, observation, and perception of bird–window collisions in Ottawa residents, 2) explore willingness to take action to reduce bird–window collisions and how this is influenced by knowledge, observation, and perception of bird–window collisions, and 3) explore barriers to taking action to reduce window collisions and potential solutions to these barriers. We discuss ways that knowledge of these barriers can inform broader campaigns for residents to treat windows.

## Methods

Data collection for this project was conducted in the fall of 2023 through an undergraduate social science course – SOCI/ANTH2180A: Foundations in Community Engagement in the Department of Sociology and Anthropology at Carleton University in Ottawa, Canada. Ethics were approved by Carleton University Research Ethics Board-B (CUREB-B #119817) on July 27, 2023.

### Study area

Ottawa is an important stopover site for migrating birds, bordering the Great Lakes-St. Lawrence lowlands. It is among the most heavily urbanized regions of Ontario, representing critical habitat for many species of conservation concern in Canada [21]. There is an active local community science program, Safe Wings Ottawa, that collects bird–windowcollision data and undertakes community outreach. Safe Wings estimates that approximately 250,000 birds die from colliding with windows each year in Ottawa [18]. An Ottawa network, Community Associations for Environmental Sustainability (CAFES), has engaged with this issue in the past. Through CAFES, four community associations were involved with this study: Fisher Heights and Area Community Association, the Glebe Community Association, the Hintonburg Community Association, and the Westboro Community Association.

### Data collection

To address our research questions, DC, instructor of SOCI/ANTH2180A, designed a survey over several months with iterative engagement with Safe Wings Ottawa, CAFES, and RTB. The survey was created using Qualtrics (Qualtrics, Provo, UT) [22]. We were rigorous in limiting the length of the survey to an estimated five minutes to reduce non-response bias. Undergraduate students were divided into four groups of eight to ten with each group working with one of four community associations to distribute the survey. One or two representatives from each community association supported their student group in the creation of unique survey recruitment materials focused on the needs and opportunities in their community, including emailing QR codes, advertisements in the community, and tabling at community centres, coffee shops, and other venues. Responses were collected between September 21 – October 30, 2023. We limited survey responses to community members that were at least 16 years of age and lived in Ottawa. All respondents were provided a question to confirm their informed consent prior to participating in the survey (S1 Survey). Respondents could answer with "I consent" or "I do not consent". Minors (those older than 16 and under 18 years) were able to consent for themselves and our ethics waived the need for a parent or guardian's consent. Upon completion of the survey, respondents were invited to a workshop

hosted by CAFES to further learn about bird–window collisions and receive a free starter kit of materials to treat their windows at home.

We were aware that members of the community association represent a group that is actively engaged in taking community action on environmental and other community issues and thus may be more inclined to indicate that they are willing to act on this issue. They also represent a demographic that is more likely to be interested in environmental action. Thus, we ensured that while we used the communication tools available to contact these groups, we also reached beyond them in our recruitment activities. Actions included email contact to each association's mailing list of active members, as well as activities such as information tables – with opportunities to complete the survey in the moment – in public gathering places, such as shopping areas or community centres.

The survey consisted of 22 questions to collect information on home traits, demographics (optional), experience with bird–window collisions (including whether respondents had knowledge of the issue, observed a bird–window collision, or perceive bird–window collisions to be an issue), willingness to act to prevent bird–window collisions, and barriers and solutions to acting. Most questions were closed on a 5-point Likert scale from "Definitely yes" to "Definitely no". Knowledge of the issue of bird–window collisions was determined with the question: "Have you ever heard about birds colliding with the windows of homes in Ottawa or not?". Observation of bird–window collisions was determined with the question: "Have you ever observed a bird colliding with a window in your current home or not?". Perception of bird–window collisions as an issue was determined with the question: "Do you think bird–window collisions at homes in Ottawa are an issue or not?". These three factors (knowledge, observation, and perception) were considered together as "awareness" to investigate the scope of people's understanding of bird–window collisions. Care was taken to ensure that the questions did not give participants the perception of a preferred answer to reduce social desirability bias. For example, the question addressing knowledge about the issue asked "Have you ever heard about birds colliding with the windows of homes in Ottawa *or not*?".

Willingness to prevent bird–window collisions by treating windows was determined with the question: "Are you willing or not willing to apply materials to the outside of your windows to reduce bird–window collisions?". Respondents answered on a Likert scale from "Definitely willing" to "Definitely not willing", with an additional option for "Already have [acted to address bird–window collisions]". To assess barriers to taking action to prevent bird–window collisions we asked, "What stops, or would stop you from taking action (or more action)?" and allowed respondents to choose as many barriers from a list of ten (e.g., time/getting around to it; cost) and/or "other" with the opportunity to specify in a text box. Finally, to explore solutions to barriers preventing action we asked, "What would support you to take action?" and allowed respondents to select from a list of seven (e.g., someone to come and apply the materials; free materials) and/or "other" with the opportunity to specify in a text box. The full survey can be found in the supplementary material (S1 Survey). The list of barriers and solutions to preventing bird–window collisions were generated by experts at Safe Wings Ottawa and through a review of the literature [12,13,15,17].

### Data analysis

All analyses were performed in R statistical software (v.2024.12.0; R Core Team, 2024) [23], and figures were made using the *ggplot2* package [24]. Responses that included answers to all questions barring the optional demographic questions were considered complete and used for analysis. We assessed the correlation between knowledge (i.e., hearing about) of bird–window collisions, observation (i.e., seeing) of bird–window collisions, perception of bird–window collisions as an issue, and willingness to act on bird–window collisions using Spearman's correlation coefficient using the *Hmisc* package [25] given the ordinal nature of the data. We removed those who selected "already have [acted to address bird–window collisions]" under the "willingness" category (n = 32), resulting in a total of 390 responses used for analysis. The resulting correlation matrix figure was created using *ggcorrplot* [26].

To explore the relationship between willingness and the frequency of each barrier, we fit a cumulative link mixed effect model with Laplace approximation, using the *ordinal* package [27]. We included willingness as the response variable with the absence or presence of a barrier, age, gender, housing type, and ownership of a birdfeeder as covariates, and years spent in Ottawa as a random effect. Only respondents with answers to all demographic questions were included (n = 355; 84% of complete responses). Two response types for gender (i.e., "Transgender" and "Non-binary") had too few responses for analysis and were removed. Those who responded "already have" under the "willingness" category were also removed, resulting in a total of 311 responses used for this analysis. Forest plots were created to depict the associations of each variable on the response as parameter estimates (PE) with 95% confidence intervals. A Spearman's correlation was conducted to test for multicollinearity prior to building the model; no covariates had correlation coefficients over 0.5.

To determine which barriers correlated most closely with different solutions selected by respondents, we calculated the Cramer's V correlation coefficient using the *vcd* package [28]. Cramer's V was determined to be the best correlation test for this scenario, as it works for nominal or binary variables that form more complex (i.e., greater than 2x2) contingency tables [29]. All de-identified data and code is available on an Open Science Framework repository DOI: https://doi.org/10.17605/OSF.IO/GBHCF.

## Results

We received a total of 465 survey responses, of which 422 were complete and included in further analysis. The majority (91.5%) of respondents had lived in Ottawa for at least one year, with 62.1% having lived in Ottawa for over five years. The largest proportion of respondents were women (74.3% of all respondents) and people who identify as ethnically white (85.7% of all respondents). The most responsive age group was older adults (50+) (37.3%), followed by those aged 31–50 (33.2%). Most residents (63.0%) lived in a two to four storey house, followed by a one storey house (16.1%), an apartment with greater than six storeys (12.8%), or an apartment with fewer than six storeys (8.1%). About a third of respondents (35.1%) had a bird feeder. Most survey respondents (66.6%) lived in the four core neighborhoods where surveys were distributed, the other 33.4% lived in a range of neighborhoods across Ottawa.

### Awareness of bird–window collisions

Most survey respondents were aware of bird–window collisions, had previous knowledge of bird–window collisions and perceived it to be an issue to some extent. 91.0% of respondents had previous knowledge of bird–window collisions and 87.9% perceive window collisions an issue in Ottawa despite only 57.3% of respondents having observed a window collision before (Fig 1).

We investigated correlations between knowledge, observation, and perception of respondents to address bird–window collisions at their residences (Fig 2). Previous knowledge of window collisions was correlated with perception of window collisions as an issue (r = 0.36, p < 0.01). However, more than half of respondents that had not heard of window collisions still perceived them to be an issue (60.5%). Knowledge of window collisions was also correlated with observation of a bird–window collision (r = 0.31, p < 0.01).

### Willingness to address bird–window collisions

Most survey respondents were willing to treat their windows (79.6%) and only 7.6% of respondents identified they had already treated their windows. Of these respondents, 91.8% of them also considered bird–window collisions to be an issue and 58.7% had observed a bird–window collision before. Of respondents who were not willing to treat their windows, 16.7% had not heard of bird–window collisions, 51.8% had not observed a bird–window collision, and 38.9% do not think bird–window collisions are an issue in Ottawa.

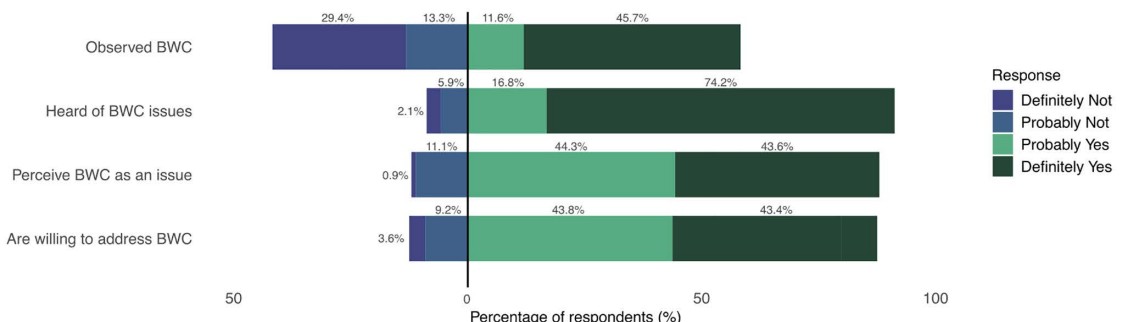

**Fig 1. Proportions of 422 survey responses "definitely not", "probably not", "probably yes", and "definitely yes" to questions corresponding to observation, knowledge, perception and willingness to address bird–window collisions in Ottawa, Canada.** Those who responded "already have" for willingness to address bird–window collisions were grouped with "definitely yes". Negative values (i.e., values to the left of zero) represent the percentage of respondents who selected "definitely not" or "probably not". Positive values represent "probably yes", "definitely yes" or "already have".

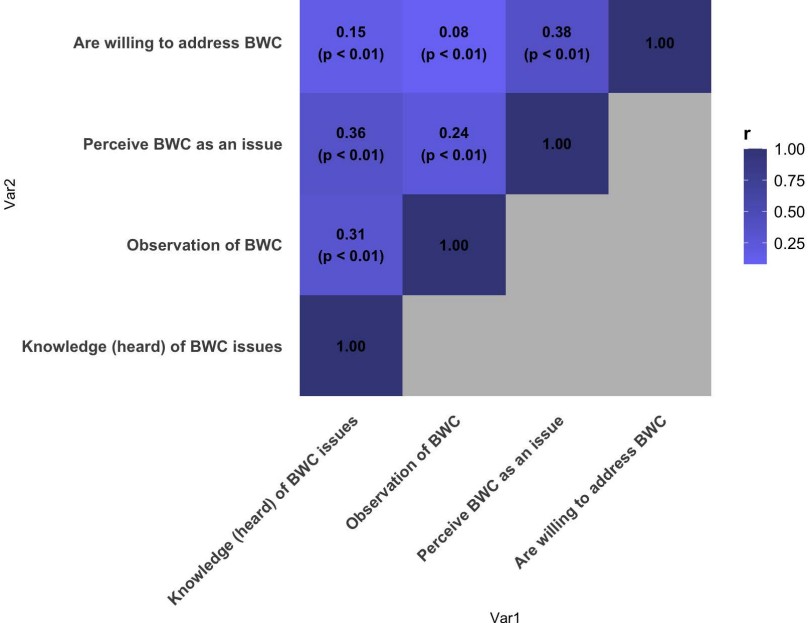

**Fig 2. Spearman's rank-order correlation test results for respondents' knowledge (i.e., whether respondents had heard of bird–window collision issues), observation (i.e., whether respondents have observed a bird–window collision), perception (i.e., whether they perceive bird–window collision to be an issue) and willingness (i.e., willingness to act on bird–window collision issues) in Ottawa, Canada.** Responses were ranked on a Likert scale from 1 to 4 from "definitely no" to "definitely yes". Those who responded to "already have [acted on bird-window collision issues]" under the "willingness" category were removed, resulting in a total of 390 out of 422 complete responses used for analysis.

Perception of bird–window collisions as an issue had the strongest relationship to willingness (Fig 2) (r = 0.38; p < 0.01). Knowledge of window collisions was only weakly related to willingness (r = 0.15; p < 0.01). Of survey respondents who were knowledgeable about window collisions, 88.3% were willing or had already treated their windows. However, of those that were not knowledgeable, 76.1% were still willing to treat their windows. Previous observation of a collision had the weakest relationship to willingness (r = 0.08; p < 0.01). However, of those that had not seen a window collision, 84.4% were still willing or had already taken action to treat their windows. Of those who had seen a window collision, 89.3% were willing or had already taken action to treat their windows.

Demographic variables were also associated with willingness. Our cumulative link mixed model (Fig 3, numerical results in S1 Table) found that respondents who selected "Male" as their gender were significantly less willing to treat windows than those who selected "Female" (PE±SE=−1.10±0.29, p<0.01). Compared to respondents aged 50+, those aged 31−50 were less willing to treat their windows, although not significantly (PE±SE=−0.30±0.30, p=0.32), and those aged 16−30 were significantly less willing to treat their windows (PE±SE=−0.85±0.36, p=0.02). Owning a birdfeeder also affected willingness, where respondents without a birdfeeder were significantly less willing to treat their windows (PE±SE=−0.69±0.31, p=0.02) than respondents who owned a birdfeeder. Participants whose neighbours owned a birdfeeder, while they themselves did not, were even less willing to treat their windows (PE±SE=−0.93±0.35, p<0.01). When it came to housing type, respondents who resided in a one storey house demonstrated a similar amount of willingness to those who resided in a two-four storey house. Those who resided in apartments over six storeys were slightly less willing to treat their windows, although not significantly (PE±SE=−0.61±0.38, p=0.11). However, those who resided in apartments under six storeys were significantly more willing to treat their windows (PE±SE=1.20±0.54, p=0.02).

## Barriers to action

For all respondents, the most identified barrier to treating windows was that birds infrequently or never collide with their windows (36.7%), followed by time (35.5%). Other commonly identified barriers included wanting to have a clear view from windows (29.4%), cost (27.0%) and access to the outside of windows to apply materials (27.0%) (Fig 4).

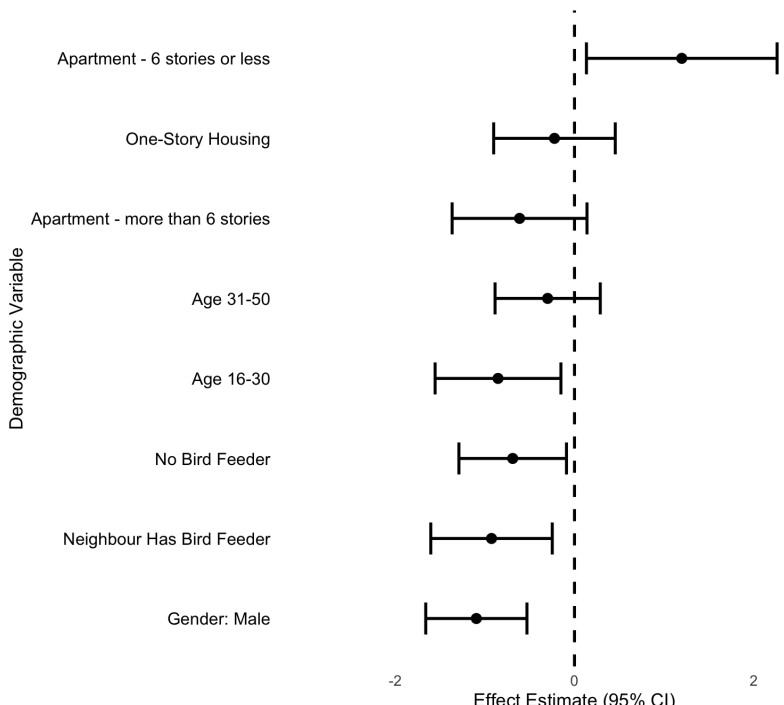

**Fig 3. Forest plot of results from cumulative link mixed model of relationship between willingness and age, housing type, gender, and ownership of a bird feeder from survey respondents from Ottawa, Canada.** Reference categories for age, housing type, gender, and ownership of a bird feeder are "over 50", "2-4 storey house", "Female", and "Owns a birdfeeder", respectively. Points depict parameter estimates with error bars depicting 95% confidence intervals. Estimates with confidence intervals that do not overlap with zero are significant. Positive estimates indicate the barrier is correlated with willingness, while negative estimates indicate the barrier is correlated with unwillingness.

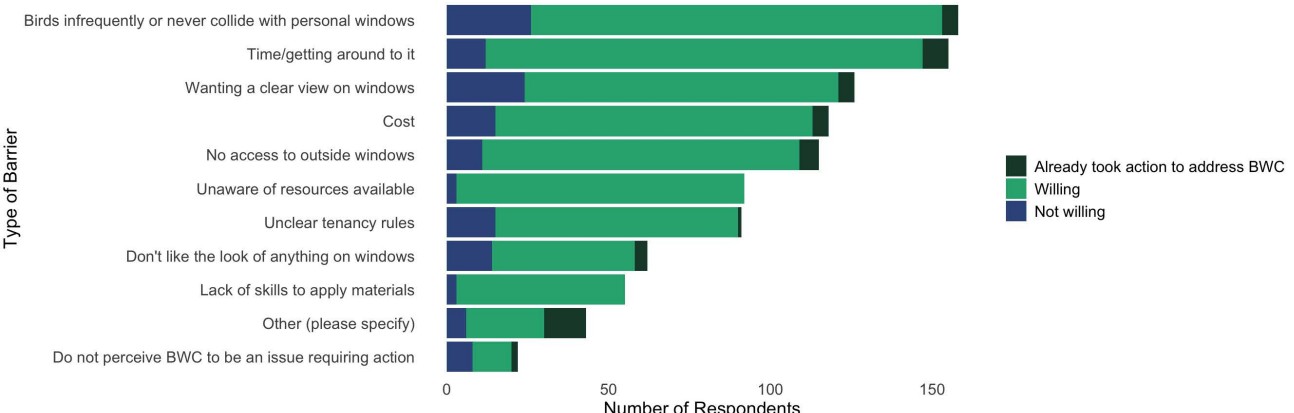

**Fig 4. Number of respondents who selected each barrier to taking action to prevent bird–window collisions at their residence.** For respondents who identified each barrier, the total number that also reported/answered they were not willing, willing, or have already acted to address bird–window collisions.

422 respondents from Ottawa, Canada. Multiple barriers could be selected by each respondent. Respondents who said "probably not willing" and "definitely not willing" are grouped into "not willing". Respondents who said "probably willing" and "definitely willing" are grouped into "willing".

For respondents that are willing to treat their windows, time was the largest barrier (39.0%) and the perception that birds infrequently collide with their windows was the second most identified (37.2%). For respondents willing to treat their windows, only 3.3% identified they did not think bird–window collisions were an issue requiring action. For those not willing to treat their windows, birds infrequently or never colliding with their windows was the most identified barrier (48.2%), followed by wanting to have a clear view from their windows (44.4%). Cost, unclear tenancy rules, and not liking the look of anything on their windows were other commonly identified barriers by those not willing to treat their windows.

While birds infrequently colliding with respondents' windows and lack of time were the most commonly selected barriers, they did not have a significant influence on willingness. Barriers with significant negative effects on willingness were aesthetics (those not liking the look of anything on their windows) (PE±SE = −0.86 ± 0.36, p = 0.02), not perceiving window collisions an issue requiring action (PE±SE = −1.60 ± 0.60, p < 0.01), and wanting a clear view from their windows (PE±SE = −0.78 ± 0.29, p < 0.01). Model results suggest these barriers are negatively related to willingness, where respondents that value aesthetics, wanting a clear view from their windows, and not perceiving this an issue needing action are significantly less willing to take action to treat their windows (Fig 5, numerical results in S2 Table).

## Solutions to barriers

Amongst all respondents, providing free materials was the most commonly selected way to support people treating their windows (46.6%) (Fig 6). Aesthetically pleasing materials (33.6%, selected as "Application of materials I like the look of"), being given clear instructions (29.7%), and having someone else apply materials for them (28.9%) were also identified as strong sources of support. For those willing to treat their windows, more information on treatment options was also important (31.2%). For those not willing to treat their windows, evidence of the need to take action was the most frequently identified solution to support action (53.8%), followed by free materials (32.7%) and aesthetically pleasing materials (32.7%).

We found that many of the strongest correlations between barriers to action and solution to take action for bird–window collisions identified by respondents were those where the presented solution directly addressed a specific barrier (e.g., unclear tenancy rules and receiving landlord permission, Cramer's V = 0.8; wanting a clear view on windows and

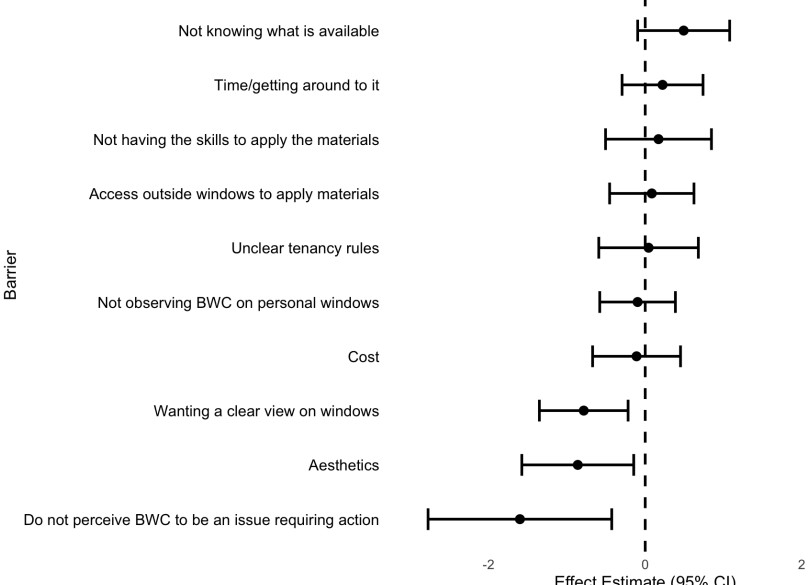

**Fig 5. Forest plot of results from cumulative link mixed model of relationship between willingness and barriers selected by respondents from Ottawa, Canada.** Points depict parameter estimates with error bars depicting 95% confidence interval. Positive estimates indicate the barrier is correlated with willingness, while negative estimates indicate the barrier is correlated with unwillingness. Estimates with confidence intervals that do not overlap with zero are significant.

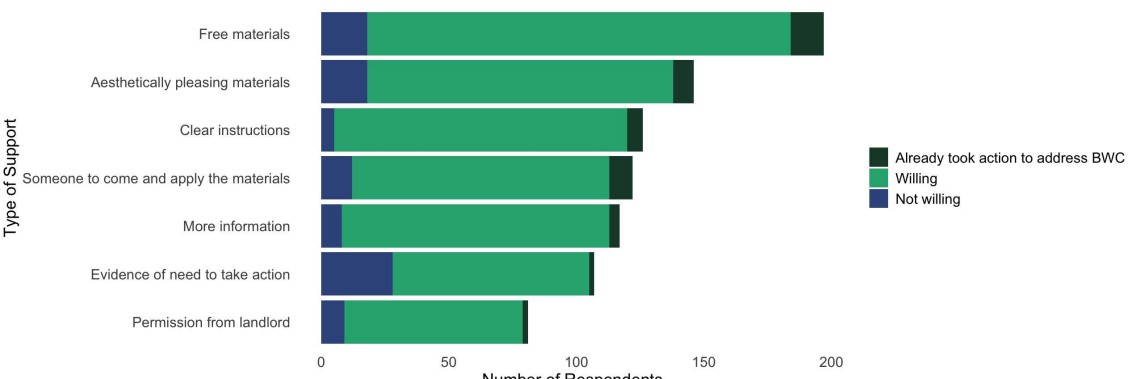

**Fig 6. Types of solutions selected by 422 respondents from Ottawa, Canada.** Responses are separated by those who are not willing, willing, and already have acted to address bird–window collisions. Respondents who said "probably not willing" and "definitely not willing" are grouped into "not willing". Respondents who said "probably willing" and "definitely willing" are grouped into "willing". Multiple solutions could be selected by each respondent.

aesthetically pleasing materials, Cramer's V = 0.42) (Fig 7). Other strong correlations (Cramer's V > 0.25) [29] included lack of awareness of available resources and clear instructions (Cramer's V = 0.35), and lack of time and provision of free materials (Cramer's V = 0.39).

## Discussion

Given high bird mortality caused by window collisions at residential homes, encouraging residents to treat their windows is an important bird conservation intervention. As such, understanding factors influencing people's behaviours in treating

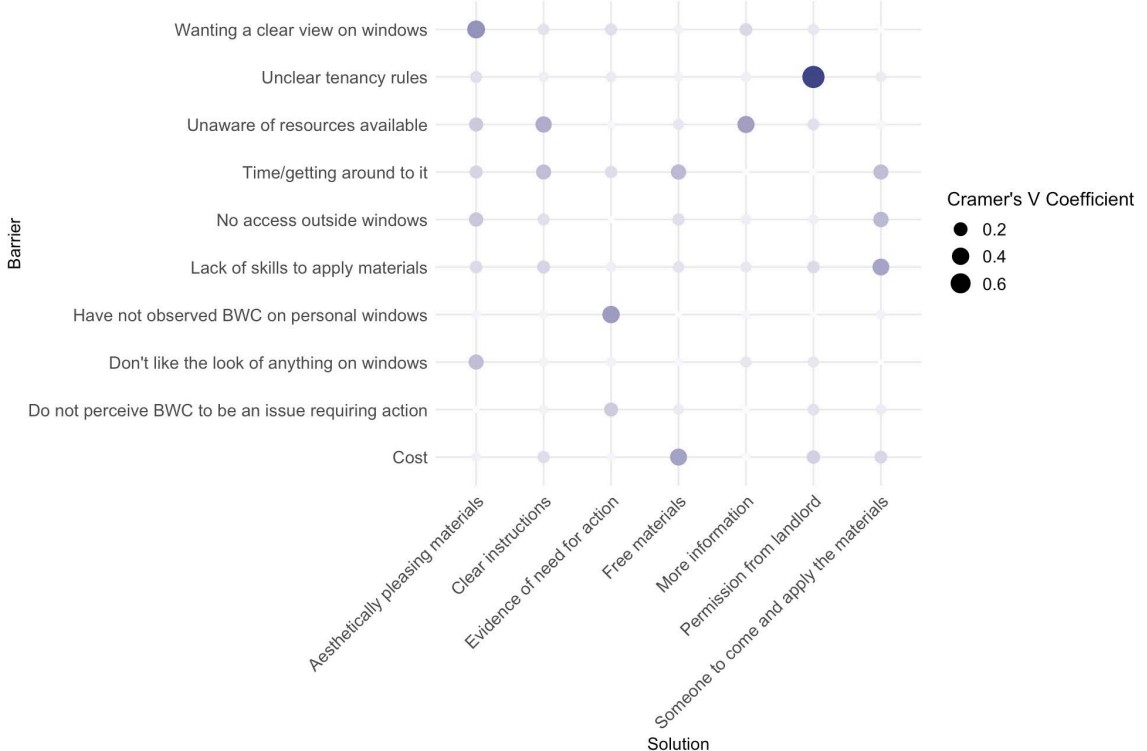

**Fig 7. Cramer's V correlation coefficient showing which solutions are most often associated with barriers identified by 422 respondents from Ottawa, Canada.** Respondents were able to select multiple barriers and solutions each.

their windows can help overcome barriers and mobilize action. Of our survey respondents, most were knowledgeable about bird–window collisions, have not yet treated their windows, but indicated they were willing to treat their windows. Despite overall awareness and willingness, very few respondents had already treated their windows. Key barriers respondents identified as preventing them from treating windows were the perception that birds infrequently collide with their windows and a lack of time, and free and aesthetically pleasing materials with clear application instructions were potential solutions. Additionally, despite many respondents having previously heard of window collisions and perceiving it to be an issue, many identified they need more evidence that this is an issue requiring action.

## Awareness and willingness to take action to prevent bird–window collisions

Pro-environmental behaviours are influenced by personality, attitudes, knowledge of the issue, potential actions, and perceptions or knowledge of behaviour efficacy [20,30]. Motivation and willingness are also influenced by socioeconomic and demographic context [31]. We found that respondents identifying as female and those over the age of 31, and those with bird feeders were significantly more likely to be willing to treat their windows. This is supported by other studies which have found that those more likely to engage with pro-environmental behaviours are typically older, white females [12,32,33] and those who engage with bird-related nature recreation activities [33].

Typically, efforts to share knowledge of an issue are done under the assumption that being knowledgeable makes people more likely to act [19]. However, the strength of this relationship varies among different types of pro-environmental behaviours [34], and in our study, we did not find knowledge of bird–window collisions to be a factor in residents treating their windows. We found that some people perceived window collisions as an issue without having

previous knowledge, suggesting that perception is more influential in willingness to take action, which has been similarly found with other pro-environmental behaviours related to homeowners and pollinators [32]. It is possible that people do not need to be knowledgeable about specific bird-conservation issues to have a sense of caring or responsibility to protect birds for their inherent value [17], and this sense of responsibility or obligation to act additionally influence the desire for an individual to engage with pro-environmental behaviours [20]. However, survey responses of observing a window collision at home had the weakest correlation with willingness to address window collisions suggesting that for survey respondents, a sense of personal responsibility or obligation may not arise from observing a collision. Knowledge was moderately correlated with observation of a window collision, but it is difficult to discern if people who already know about collisions are then more likely to have observed one or if observation of a collision elicits learning of the issue. While there may be little to link environmental knowledge with behaviours [34], it is possible different levels or types of knowledge of an issue can influence behaviours. In our study, many respondents were generally willing to treat their windows and identified the need for more information or evidence this is an issue at their homes, suggesting that increasing knowledge of bird–window collisions to describe how this is an issue at homes and potential solutions and efficacies of solutions may further improve their willingness to take action at their homes.

### Barriers to action and potential solutions

The most common barrier to taking action to prevent bird–window collisions amongst all respondents was the perception that birds infrequently or never collide with their windows. However, there was no significant relationship between this perception and the respondents' willingness to treat windows. This may indicate that for those who are willing to treat their windows, the issue may not be considered pressing enough at their own homes to warrant the cost, time, or effort to take action. This may again relate to the fact that even if residents are knowledgeable about the issue of bird–window collisions, they may not understand the magnitude of the issue, particularly at residences [15,17]. While many residents believe that birds infrequently or never collide with their windows, this may not reflect the true rates of collisions at homes, as most residential collisions are not witnessed [35]. Collisions are a sudden event often followed by predation of the carcass, and can be easily missed, especially given that many people are not in their homes for most of the day. Additionally, many homeowners have no reason to be looking for evidence of collisions, and if they are, many collisions do not result in visible evidence to find like a carcass or feathers on a window [13,35,36]. Messaging strategies aiming to communicate the problem of collisions at homes and the potential for residents to contribute to solutions could potentially help overcome this barrier.

Current messaging and community science programs aiming to prevent bird–window collisions tend to focus on high-rise commercial buildings as primary sources of window collisions and targets for mitigation efforts as they account for more collisions per building than other building types [2,37]. In Ontario, some municipal bird-safe guidelines have begun to address this problem when constructing new commercial or industrial buildings, but it is difficult to regulate bird-safe design at homes [38]. Commercial buildings can present high-profile cases like McCormick Place in Chicago, where in fall of 2023, nearly 1,000 birds died overnight from colliding with McCormick Place and this event garnered public support that was successful in treating windows of this building [6]. However, annual Canadian estimates of collision mortalities attribute only ~11% of all collisions to commercial low- and high-rise buildings in comparison to houses, which account for 90% of annual collisions [2]. While these high-profile cases are important in raising awareness to prevent collisions, they may lead to an under-emphasis on the role of houses and low-rise buildings in contributing to bird–window collisions. This may lead to people believing bird–window collisions are not particularly an issue at homes and addressing the issue is more the responsibility of others [17]. Further research could assess people's breadth of knowledge on the issue of window collisions at home versus other buildings to explore if this is driving the gap between willingness and taking action to prevent bird–window collisions for those who are knowledgeable.

Personal aesthetic preferences were also a commonly identified barrier, where many wanted a clear view from windows and identified aesthetically pleasing materials as a solution. Previous studies have also found that homeowners value unobstructed views and are less willing to retrofit their windows if glass aesthetics are affected [12,15,39]. This may be influenced by social norms and practices, or lack there-of, around bird-friendly window treatments. As they are not common on residential buildings, there may be little social pressure to implement window treatments. However, while people value clear views from their windows, Sheng et al. [39] noted that survey respondents would be willing to compromise the view if they knew the designs would prevent bird fatalities.

Practical barriers were also identified by respondents as preventing them from treating their windows. Time was the most common, but cost, access, and tenancy rules were also identified as barriers by those willing to treat their windows. Practical solutions addressing these issues included communicating clear instructions for window treatment options that are low cost and low time investment or providing free materials and/or services to treat windows for respondents to tackle time, access, or ability barriers (Fig 4). In general, our survey group represents people that would be more willing to engage with pro-environmental behaviours. Other studies have found that a similar demographic is also more willing to pay for environmental solutions [12,40] but cost still presented a barrier for respondents in treating their windows. Along with provision of free materials as a top solution to overcome this barrier, most survey respondents communicated they would be willing to apply window treatments if a free kit was given to them. Financial aid is a broader solution to help address this problem, for example, monetary incentives [12] or government subsidies [17], and if used in conjunction with products that are easily applied with clear and simple instructions, this could present an opportunity to more individuals treating their windows.

## Limitations and future directions

Our survey focused on four specific neighbourhoods in Ottawa, with some broader participation around the city. Many responses came from residents that participate with a community association and may be more likely to be interested in taking action regarding environmental issues than other residents. Thus, applying our findings to the entire city or more broadly to other cities in Canada should be done with caution. However, our survey provides one of the first to our knowledge to explore the relationships between awareness, willingness, and barriers and solutions for home residents in Canada to make their windows bird-friendly and, with over 400 responses, offers valuable insights.

We took a broad approach to explore barriers to taking action to prevent bird–window collisions. As such, we did not assess more detailed influences of different factors on responses. For instance, we did not assess the type of knowledge people had of window collisions at their homes, nor did we assess their knowledge of window collision treatments. Additionally, the only preventative action we explored was window treatments rather than including actions such as relocating bird feeders or altering landscaping at homes [12,14], advocating for bird-friendly guidelines in cities, or donating to organizations that work with window collisions. These would make interesting avenues for further research to motivate action to combat bird–window collisions.

## Potential pathways and opportunities for increasing action

While our results indicate the importance of knowledge in influencing willingness to take action on bird–window collisions, they also suggest that a shift in the type of knowledge presented by conservation organizations is needed to emphasize the importance of residential homes in preventing collisions. We found that respondents were willing to address this issue but required more information, particularly that this is an issue at their homes, suggesting a perception that individual actions at home might not be impactful. This is useful for collision advocacy groups to better communicate the prevalence of bird–window collisions at low-rise buildings and homes. Targeted messaging should emphasize that individual actions of treating windows at homes with effective treatments can prevent collisions and will have a large cumulative impact on reducing bird–window collisions at local, regional, and national scales. This messaging could contribute to overcoming

barriers related to aesthetics and values related to the view from windows if people felt the outcomes of their actions outweighed these barriers (as indicated by Sheng et al. [39]). Additionally, providing information of different types of window treatments and using specific supporting evidence that these reduce collisions has the potential to improve people's behaviours in treating their windows, such as the information presented on the FLAP Canada webpage "Stop Birds from Hitting Windows" [41] or the American Bird Conservancy's Products and Solutions Database [42]. To address practical barriers, another potential opportunity is for advocacy groups to help make treating windows at home as easy as possible for residents (e.g., identifying window washing companies that treat windows and providing volunteer services to treat windows, e.g., [43]). Efforts to make treating windows an easier process to accomplish, aesthetically pleasing, and connected to peoples' values and sense of responsibility, would help bridge the gap between willingness and taking action to prevent bird–window collisions.

## Supporting information

**S1 Survey. Qualtrics survey questionnaire on bird–window collisions for residents of Ottawa, Canada.**
(DOCX)

**S1 Table. Results from cumulative link mixed model of relationship between willingness and age, housing type, gender, and ownership of a bird feeder.**
(DOCX)

**S2 Table. Results from cumulative link mixed model of relationship between barriers and willingness.**
(DOCX)

## Acknowledgments

We would like to thank A. Keller-Herzog, P. MacDonald, and others at the Community Associations for Environmental Sustainability, J. Niwa and W. English at Safe Wings Ottawa, and D. Doherty at Bird-friendly Ottawa for supporting this project. We thank community association representatives W. Burpee and P. MacDonald (FHACA), J. Freeman and T. Beauchamp (CGA), R. Philar (HCA), and D. Chapman (WCA), and the SOCI/ANTH 2180 class for recruiting survey respondents.

## Author contributions

**Conceptualization:** Deborah Conners, Rachel T. Buxton.

**Data curation:** Aalia I. Khan, Deborah Conners, Rachel T. Buxton.

**Formal analysis:** Aalia I. Khan.

**Funding acquisition:** Rachel T. Buxton.

**Investigation:** Deborah Conners, Rachel T. Buxton.

**Methodology:** Aalia I. Khan, Deborah Conners, Rachel T. Buxton.

**Project administration:** Deborah Conners, Rachel T. Buxton.

**Resources:** Deborah Conners, Rachel T. Buxton.

**Supervision:** Deborah Conners, Rachel T. Buxton.

**Visualization:** Aalia I. Khan.

**Writing – original draft:** Anastasia J. V. Lysyk.

**Writing – review & editing:** Anastasia J. V. Lysyk, Aalia I. Khan, Deborah Conners, Rachel T. Buxton.

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
