## [Decision Letter · Decision Letter 0]

29 Aug 2025

Dear Dr. Lysyk,

Thank you for submitting your manuscript to PLOS ONE. After careful consideration, we feel that it has merit but does not fully meet PLOS ONE’s publication criteria as it currently stands. Therefore, we invite you to submit a revised version of the manuscript that addresses the points raised during the review process.

We look forward to receiving your revised manuscript.

Kind regards,

Teddy Lazebnik

Academic Editor

PLOS ONE

Journal Requirements:

[We would like to thank A. Keller-Herzog, P. MacDonald, and others at the Community Associations for Environmental Sustainability, J. Niwa and W. English at Safe Wings Ottawa, and D. Doherty at Bird-friendly Ottawa for supporting this project. We thank community association representatives W. Burpee and P. MacDonald (FHACA), J. Freeman and T. Beauchamp (CGA), R. Philar (HCA), and D. Chapman (WCA), and the SOCI/ANTH 2180 class for recruiting survey respondents. RTB, AJVL, and AK were funded by NSERC (RGPIN 04888), the Kenneth Molson Foundation, and Environment and Climate Change Canada (GCXE24S042).]

[Funding was provided to RTB, AJVL, and AK through three funding sources.

1) Natural Science and Engineering Research Council of Canada grant (RGPIN 04888) (https://www.nserc-crsng.gc.ca/index_eng.asp).

2) the Kenneth Molson Foundation (https://fondationmolson.org/en/).

3) Environment and Climate Change Canada (GCXE24S042) (https://www.canada.ca/en/environment-climate-change.html).

Funders did not play any role in study design, data collection and analysis, decision to publish, or preparation of the manuscript.]

5. Please note that PLOS One has specific guidelines on code sharing for submissions in which author-generated code underpins the findings in the manuscript. In these cases, we expect all author-generated code to be made available without restrictions upon publication of the work. Please review our guidelines at https://journals.plos.org/plosone/s/materials-and-software-sharing#loc-sharing-code and ensure that your code is shared in a way that follows best practice and facilitates reproducibility and reuse.

6. Please provide additional details regarding participant consent. In the ethics statement in the Methods and online submission information, please ensure that you have specified (1) whether consent was informed and (2) what type you obtained (for instance, written or verbal, and if verbal, how it was documented and witnessed). If your study included minors, state whether you obtained consent from parents or guardians. If the need for consent was waived by the ethics committee, please include this information.

7. You indicated that you had ethical approval for your study. In your Methods section, please ensure you have also stated whether you obtained consent from parents or guardians of the minors included in the study or whether the research ethics committee or IRB specifically waived the need for their consent.

Additional Editor Comments:

Based on the reviewers' comments, I invite the authors to submit a major revision of the manuscript.

Reviewers' comments:

Reviewer's Responses to Questions

**Comments to the Author**

1. Is the manuscript technically sound, and do the data support the conclusions?

Reviewer #1: Yes

Reviewer #2: Yes

2. Has the statistical analysis been performed appropriately and rigorously?

Reviewer #1: Yes

Reviewer #2: Yes

3. Have the authors made all data underlying the findings in their manuscript fully available?

Reviewer #1: No

Reviewer #2: Yes

4. Is the manuscript presented in an intelligible fashion and written in standard English?

Reviewer #1: Yes

Reviewer #2: Yes

Reviewer #1: This article summarizes a survey conducted in Ottawa, Canada, about bird-window collisions, which are a major cause of bird deaths in North America. The survey found that most Ottawa residents are aware of and concerned about bird-window collisions, and a high percentage are willing to treat their windows to prevent them. However, several factors act as barriers to implementing window treatments, especially the perception of infrequent collision and aesthetic concerns about how window treatments look and affect clear views as well as lack of time and costs associated with treating windows. The study suggests that offering free materials, aesthetically pleasing options, and clear instructions could help overcome these barriers. Ultimately, the authors advocate for targeted public messaging to highlight the impact of residential homes on this issue and to encourage widespread adoption of bird-friendly window treatments.

The article is interesting, well written and succinct. I provide comments below to help clarify the writing and the presentation of results and subsequent discussion.

Major comments:

- Are the legends correct on Figures 4 and 6? I ask because I don’t understand the “already have” for many of the responses such as “no access” (+ others) and what that has to do with willingness. Also, in the methods, I thought you asked the questions on a 5-pt Likert Scale but the answers aren’t presented that way. Could you please clarify all this where needed.

- Paragraph that starts at L340. There is quite a bit of literature about the gap between environmental knowledge and pro-environmental action. I suggest integrating some of that literature into this paragraph or a new one so that you can tie your results to theoretical models of this “gap”. This is one key paper along those lines: https://www.tandfonline.com/doi/abs/10.1080/13504620220145401 but there are others.

- Are the mixed models fitted with all the respondents or just the ones who do not already have treatments on their windows? If the former, when you run the models with the subset, do the results change? What about for the correlations?

- In the methods, you say that socio-economic and demographic questions were optional on the survey, what is the effective response rate for those questions, and how many respondents are included in the statistical models you build later? I believe this should be included at the very least in the figure captions, maybe as n=XXX).

- L183, will need to be changed to the link to the OSF repository upon publication

- The references are not formatted properly (e.g. journal titles should be italicized) and some references are missing information (e.g. 12 and 21). Please check those thoroughly or use a formatting software such as Zotero (free).

Minor comments:

- For future surveys you might conduct, Dillman et al. (https://sesrc.wsu.edu/about/total-design-method/) recommends writing questions as statements and then using the standard definitely agree to disagree Likert scale. So your questions about “Are you willing to not willing to X”, become “I am willing to X”.

- For Figure 2, in the grey squares, I would put the p-values or have some way to visualize them. This might work for you but is not the only way to do this: https://indrajeetpatil.github.io/ggstatsplot/articles/web_only/ggcorrmat.html You could also just add the p-value in parenthesis in the colored box for ease of interpretation (or present this Figure as a table which may be more efficient)

- Line 67, could you specify what a low-rise building is?

- Line 84, missing “on”

- Line 97 at the end, missing “and”

- Line 110, in my field, the word data is treated as a plural (datum is the singular) so it would be “data were” but I’ve noticed this seems to be journal/field specific.

- Line 123, spell out “4” as “four”. I think anything below 12 is expected to be spelled out (occurs on L 157 and 160, 192 too)

- L163, it would be helpful to have the references for the literature that informed the list of barriers and solutions (assuming it’s <~6 refs)

- L172, instead of “ran” consider “fit” and the order of the sentence could be different: you fit the model Mixed Effect Model with a cumulating link and Laplace approx..

- L212, what is the p-value for the observation and knowledge questions?

- L219-220, I don’t see that info in Figure 1.

- L219 and 225, you repeat the same info about already treated households

- L227-231 seem to be a repeat of L206-212 as well. Pick one or merge the two perhaps?

- L237, what does PE stand for?

- L235, consider replacing “also affected” by “were also associated with” since the first implies causation

- L320, consider replacing “older” with “middle-aged” or “over 50 years old”?

- L320, I think you need to rephrase that sentence of maybe replace the “and” with a “who” or “who identified with”

- Sentence starting at L322 has too many “ands”

- L325, results don’t find anything, please rephrase

- L327-328, you say multiple studies corroborate your findings, but only cite one (32). Either chance studies to study or cite more than one study.

- L332, at “Respondents”, consider making that a new paragraph.

- In a future study, it might be interesting to about observed collisions where the bird(s) died upon impact or not.

Reviewer #2: This manuscript describes survey-based research study in which a sample of citizens from various neighborhoods in Ottawa, Canada were surveyed about their knowledge and perception of bird–window collisions at residential homes, their willingness to treat windows to prevent collisions, and barriers to action. The survey resulted in several significant correlations between variables, including perception of bird–window collisions and willingness to treat windows. Somewhat surprisingly, perception of collisions as a problem that should be addressed was more significantly correlated with willingness to act than experience witnessing a bird–window collision. Another important finding is that many respondents had witnessed a collision at home but needed more evidence that such collisions are a big enough problem before they would consider treating their windows. Significant barriers to respondents treating their windows at home included time, cost, and lack of clear instructions. All these points can help inform conservation groups and collision monitoring programs tailor their educational, outreach, and advocacy efforts to better communicate that most collisions occur at low-rise residential buildings rather than at tall downtown buildings that are often featured in news media as being the top cause of collisions.

Considering the great numbers of birds that die each year from building collisions in North America alone, this study has much merit, as it helps fill a gap in the published literature and should aid conservationists and communications experts working to spread awareness of and reduce bird–building collisions. As an ecologist and researcher with years of experience leading a collision monitoring program, I was excited to see this study and look forward to further studies that will build off this.

The manuscript is well-written overall, presented in logical order, and includes sound statistical analyses, useful supplementary tables, and relevant references. I recommend accepting the manuscript, pending revisions.

Major Feedback:

1) While the authors mention some limitations of their survey in the Discussion, the Methods section does not describe any steps that were taken to reduce bias in data collection. This is a serious concern, as survey results are highly sensitive to issues such as sampling bias, non-response bias, and social desirability bias. I recommend that the authors provide more detail on how the survey was designed and administered to minimize these risks (e.g., question design, question randomization, recruitment strategy, etc.). Without this information, it is difficult to evaluate the reliability of the findings.

2) For readers who are not familiar with Likert scale charts (like me), Figure 1 may be difficult to understand as presented. I recommend adding more detailed explanations of how to interpret this to avoid confusion and perhaps add percentages inside the bars. For example: explain why Fig. 1 has positive and negative percentages, why the bars are staggered, and what this means.

3) The use of CLMMs is appropriate for ordinal responses, and I appreciate that the authors ran two separate models and presented the results in forest plots. However, the basis for Figure 3 is not entirely clear. From Table S2, it appears that the demographic model used ‘>50 years’, ‘2–4 story house’, ‘female’, and ‘owns a bird feeder’ as reference categories, but this is not explained in the text. The reference categories should be stated clearly in the main text or Fig. 3 caption so readers can interpret the effect estimates in relation to the correct baseline groups.

In addition, one statistically significant covariate (age 16-30) is not included in Fig. 3, whereas two non-significant covariates (transgender, non-binary) are included. In contrast, Fig. 5 displays all predictors, regardless of significance. I recommend clarifying the rationale for which covariates are displayed in each forest plot and ensuring consistency in presentation. Otherwise, readers may incorrectly assume that omitted variables were not significant or were not tested.

Lastly, I recommend including an explanation of what the parameter estimates mean in the main text for clarity. Currently, this information is only stated in Tables S2 & S3.

4) The Discussion section is quite wordy and restates much information presented in the Results section, making it unnecessarily long and repetitive. Revising for conciseness, focusing on what the results mean and why this should matter to readers, without restating the results, would make the discussion much stronger and more digestible/useful for readers.

Minor Feedback:

Minor suggestions regarding grammatical errors, typos, etc. can be found as comments in the attached pdf. Other minor feedback is listed below.

1) Throughout the manuscript, the authors use a hyphen (-) between “bird” and “window” when referring to collisions, but an en dash (–) should be used instead. Hyphens should be used in compound adjectives, e.g. bird-friendly materials, whereas an en dash should be used to indicate linkages between two things, e.g. bird–window collisions.

2) Ln 38-40: It isn’t helpful to report parameter estimates in the abstract, as readers cannot interpret these outside the context of the full model results. I recommend highlighting the meaning of the results (relative strength, significance) rather than raw statistical values in the abstract.

3) Ln 65-66: The article cited here (Klem & Saenger 2013) isn’t about window films or decals. They tested UV-treated windows and Acopian BirdSavers in this study. There are several better options to cite here, including DeGroot et al. 2022 (doi: 10.7717/peerj.13142) or Bird-friendly Building Design from American Bird Conservancy (https://abcbirds.org/wp-content/uploads/2015/05/Bird-friendly-Building-Guide_2015.pdf).

4) Ln 182-183: Please add an accession number or URL prior to publication.

5) Ln 186-195: It appears that the percentages reported in this paragraph are based on all 465 respondents rather than the 422 included in the analysis. For clarity and consistency throughout the Results section, please base these percentages on the number of responses used in the analysis.

6) Ln 187-189: It is unclear if the percentages of older respondents and those who identified as white are based on the number of female respondents only or all respondents. Please reword to clarify.

7) Ln 189-190: It is unclear why counts are reported here rather than percentages. I recommend reporting percentages for consistency. Again, these should be based on the number of responses used in the analysis.

8) Ln 212: For consistency, please report the p-value with the correlation coefficient, as in line 209.

9) Ln 219-220, 224-225: The percentage of respondents who had already treated their windows is reported twice in this paragraph. I recommend deleting the last sentence and removing the word “only” from the first sentence.

10) Figure 3: To aid in comparing the importance of variables within categories, I recommend reordering the variables to be grouped by category rather than strictly in increasing numeric order. I also recommend denoting which variables are significant to aid readers who may be less familiar with interpreting forest plots.

11) Ln 289: Please clarify what is meant by “more information”. (More information about what?)

12) Figure 5: As with Fig. 3, I recommend denoting which barriers are significant.

13) Ln 320-328: Considering that the authors reported in the Results that the largest proportion of respondents were older, white females, I wonder if this may introduce bias into the survey results and interpretation via sampling bias. Thus, I would like to know if the authors took this into consideration.

14) Ln 333-336: It is not clear if the authors are referring to all respondents or a subset of respondents here, nor why this group is considered more likely to act. Please clarify. Also, the end of this sentence reads clumsily. I suggest revising to something like, “…suggesting barriers that prevent people from taking action still exist.”

15) Ln 362-363: I suggest changing “under” to “nearly” for clarity.

16) Ln 399: I suggest inserting “frequently” before “followed”, as carcasses are not always depredated.

17) Ln 426-427: This is true, but considering the survey results, it seems this will only lead to a significant number of individuals treating their windows if the monetary aid is in conjunction with suggestions for bird-friendly products with clear, simple instructions (such as FeatherFriendly). I feel this is worth mentioning.

18) Ln 447-450: The way this sentence reads, it suggests that such a shift has already occurred, rather than a shift is needed. I suggest revising as, “…they also suggest a shift is needed in the type…”.

19) Ln 457: Perhaps replace “a few” with “some”, as this makes it seem that very few collisions occur at individual homes.

20) Ln 461-463: An excellent resource that could be included as an example here is the Glass Collisions section of American Bird Conservancy’s website, particularly their Products & Solutions Database (https://abcbirds.org/glass-collisions/products-database/).

21) References: It would benefit readers to include DOI numbers for referenced articles.

22) Ln 510-511: Please correct the formatting for the Warren (2013) reference, as it is a Master’s Thesis.

**Do you want your identity to be public for this peer review?** For information about this choice, including consent withdrawal, please see our Privacy Policy

Reviewer #1: No

Reviewer #2: No

---

## [Author Response · Author response to Decision Letter 1]

26 Nov 2025

This information is repeated in the uploaded response to reviewers file.

Author Response to Editor

Dear Dr. Lazebnik,

Thank you for the helpful feedback on our manuscript. Please extend our thanks to the reviewers as well, if possible.

Below, we have detailed how we revised the manuscript to address each reviewer’s comments, suggestions, or concerns. Our response to each point is typed in bold and references to the text are included in italics . Line numbers match those in the clean version of the revised manuscript with each change marked in the track-changes version. In regards to the funding statement and acknowledgements, we do not wish to change the funding statement but have revised the acknowledgements in the manuscript.

We hope you will find the revised manuscript acceptable for publication, and we look forward to hearing from you.

Sincerely,

Anastasia Lysyk & Aalia Khan, and all Co-Authors.

Academic editor comments:

Response: We have made edits to meet the style requirements including maintaining sentence case in section headings, and bolding the figure titles in each figure caption.

[We would like to thank A. Keller-Herzog, P. MacDonald, and others at the Community Associations for Environmental Sustainability, J. Niwa and W. English at Safe Wings Ottawa, and D. Doherty at Bird-friendly Ottawa for supporting this project. We thank community association representatives W. Burpee and P. MacDonald (FHACA), J. Freeman and T. Beauchamp (CGA), R. Philar (HCA), and D. Chapman (WCA), and the SOCI/ANTH 2180 class for recruiting survey respondents. RTB, AJVL, and AK were funded by NSERC (RGPIN 04888), the Kenneth Molson Foundation, and Environment and Climate Change Canada (GCXE24S042).]

[Funding was provided to RTB, AJVL, and AK through three funding sources.

1) Natural Science and Engineering Research Council of Canada grant (RGPIN 04888) (https://www.nserc-crsng.gc.ca/index_eng.asp).

2) the Kenneth Molson Foundation (https://fondationmolson.org/en/).

3) Environment and Climate Change Canada (GCXE24S042) (https://www.canada.ca/en/environment-climate-change.html).

Funders did not play any role in study design, data collection and analysis, decision to publish, or preparation of the manuscript.]

Response: We have removed the funding statement from the acknowledgements section of the revised manuscript.

Response: We have updated the manuscript to include a DOI number to an online repository through Open Science Framework containing all relevant data and analyses (line 207).

5. Please note that PLOS One has specific guidelines on code sharing for submissions in which author-generated code underpins the findings in the manuscript. In these cases, we expect all author-generated code to be made available without restrictions upon publication of the work. Please review our guidelines at https://journals.plos.org/plosone/s/materials-and-software-sharing#loc-sharing-code and ensure that your code is shared in a way that follows best practice and facilitates reproducibility and reuse.

Response: Access to code sharing is included in our online repository through OSF.

6. Please provide additional details regarding participant consent. In the ethics statement in the Methods and online submission information, please ensure that you have specified (1) whether consent was informed and (2) what type you obtained (for instance, written or verbal, and if verbal, how it was documented and witnessed). If your study included minors, state whether you obtained consent from parents or guardians. If the need for consent was waived by the ethics committee, please include this information.

Response: We have described in the methods section that ethics were approved to describe informed participant consent through a survey question and all participants over the age of 16 were allowed to consent their own participation in the survey (lines 135-139).

7. You indicated that you had ethical approval for your study. In your Methods section, please ensure you have also stated whether you obtained consent from parents or guardians of the minors included in the study or whether the research ethics committee or IRB specifically waived the need for their consent.

Response: The ethical approval for this study allowed all participants over the age of 16 to confirm their own informed consent to participate in the survey (lines 135-139).

Reviewer #1: This article summarizes a survey conducted in Ottawa, Canada, about bird-window collisions, which are a major cause of bird deaths in North America. The survey found that most Ottawa residents are aware of and concerned about bird-window collisions, and a high percentage are willing to treat their windows to prevent them. However, several factors act as barriers to implementing window treatments, especially the perception of infrequent collision and aesthetic concerns about how window treatments look and affect clear views as well as lack of time and costs associated with treating windows. The study suggests that offering free materials, aesthetically pleasing options, and clear instructions could help overcome these barriers. Ultimately, the authors advocate for targeted public messaging to highlight the impact of residential homes on this issue and to encourage widespread adoption of bird-friendly window treatments.

The article is interesting, well written and succinct. I provide comments below to help clarify the writing and the presentation of results and subsequent discussion.

Response: Thank you very much for this positive feedback!

Major comments:

- Are the legends correct on Figures 4 and 6? I ask because I don’t understand the “already have” for many of the responses such as “no access” (+ others) and what that has to do with willingness. Also, in the methods, I thought you asked the questions on a 5-pt Likert Scale but the answers aren’t presented that way. Could you please clarify all this where needed.

Response: We have clarified these are three separate responses by removing “willingness” in the legend title and we have changed “already have” to “already have taken action to address BWC” in Fig.4. In the same figure, we have changed “no access outside windows” to “no access to outside windows”.

- Paragraph that starts at L340. There is quite a bit of literature about the gap between environmental knowledge and pro-environmental action. I suggest integrating some of that literature into this paragraph or a new one so that you can tie your results to theoretical models of this “gap”. This is one key paper along those lines: https://www.tandfonline.com/doi/abs/10.1080/13504620220145401 but there are others.

Response: Thank you for suggesting this paper, we have incorporated this and others in the revised paragraph starting on line 362.

- Are the mixed models fitted with all the respondents or just the ones who do not already have treatments on their windows? If the former, when you run the models with the subset, do the results change? What about for the correlations?

Response: Thank you for this suggestion. We agree with your feedback and have decided to remove respondents who already have treatments on their windows from our models and correlation analysis, as they have already bypassed the barriers we are exploring in our study.

Results do not change substantially when the models are fitted with the subset, with the exception that people aged 31-50 are no longer significantly less willing to act compared to reference (age 50+). Fig2, Fig3, Fig5, Table S2 and Table S3 have been updated with the new results.

- In the methods, you say that socio-economic and demographic questions were optional on the survey, what is the effective response rate for those questions, and how many respondents are included in the statistical models you build later? I believe this should be included at the very least in the figure captions, maybe as n=XXX).

Response: Thank you for your suggestion. We have included the response rate and number of respondents used for our statistical models in our methods section (line 195-200).

- L183, will need to be changed to the link to the OSF repository upon publication

Response: The DOI number to the OSF repository has been added to the manuscript (line 209).

- The references are not formatted properly (e.g. journal titles should be italicized) and some references are missing information (e.g. 12 and 21). Please check those thoroughly or use a formatting software such as Zotero (free).

Response: Thank you for your feedback. We have edited the references according to existing guidelines for PLOS ONE (journal titles are not italicized as the citation style follows the “Vancouver” format). We have added the missing information.

Minor comments:

- For future surveys you might conduct, Dillman et al. (https://sesrc.wsu.edu/about/total-design-method/) recommends writing questions as statements and then using the standard definitely agree to disagree Likert scale. So your questions about “Are you willing to not willing to X”, become “I am willing to X”.

Response: Thank you for the suggestion, which we will consider in future studies.

- For Figure 2, in the grey squares, I would put the p-values or have some way to visualize them. This might work for you but is not the only way to do this: https://indrajeetpatil.github.io/ggstatsplot/articles/web_only/ggcorrmat.html You could also just add the p-value in parenthesis in the colored box for ease of interpretation (or present this Figure as a table which may be more efficient)

Response: Thank you for this suggestion! We have now included p-values in the colored boxes for Figure 2.

- Line 67, could you specify what a low-rise building is?

Response: Addressed in line 66.

- Line 84, missing “on”

Response: Addressed.

- Line 97 at the end, missing “and”

Response: Addressed.

- Line 110, in my field, the word data is treated as a plural (datum is the singular) so it would be “data were” but I’ve noticed this seems to be journal/field specific.

Response: Addressed.

- Line 123, spell out “4” as “four”. I think anything below 12 is expected to be spelled out (occurs on L 157 and 160, 192 too).

Response: Changed here and throughout the paper.

- L163, it would be helpful to have the references for the literature that informed the list of barriers and solutions (assuming it’s <~6 refs)

Response: Thank you, we have included references at the end of this sentence (Line 180).

- L172, instead of “ran” consider “fit” and the order of the sentence could be different: you fit the model Mixed Effect Model with a cumulating link and Laplace approx..

Response: Changed.

- L212, what is the p-value for the observation and knowledge questions?

Response: Added to results.

- L219-220, I don’t see that info in Figure 1.

Response: Addressed - Fig1 is no longer cited for this info.

- L219 and 225, you repeat the same info about already treated households.

Response: Removed repeated line.

- L227-231 seem to be a repeat of L206-212 as well. Pick one or merge the two perhaps?

Response: Thank you for your suggestion. Results on L206-212 (now 233-238) are about correlations among knowledge, observation, and perception, while results on L227-231 (now 254-258) are about correlations with willingness. We have removed the term “willingness” (line 231-232) to make this more clear.

- L237, what does PE stand for?

Response: PE stands for “parameter estimates” - we have defined this term in the methods sectio

---

## [Decision Letter · Decision Letter 1]

21 Jan 2026

Barriers and opportunities to preventing residential bird-window collisions

PONE-D-25-29481R1

Dear Dr. Lysyk,

We’re pleased to inform you that your manuscript has been judged scientifically suitable for publication and will be formally accepted for publication once it meets all outstanding technical requirements.

Kind regards,

Teddy Lazebnik

Academic Editor

PLOS One

Additional Editor Comments (optional):

Reviewers' comments:

Reviewer's Responses to Questions

**Comments to the Author**

Reviewer #2: All comments have been addressed

2. Is the manuscript technically sound, and do the data support the conclusions?

Reviewer #2: Yes

3. Has the statistical analysis been performed appropriately and rigorously?

Reviewer #2: Yes

4. Have the authors made all data underlying the findings in their manuscript fully available?

Reviewer #2: Yes

5. Is the manuscript presented in an intelligible fashion and written in standard English?

Reviewer #2: Yes

Reviewer #2: The authors did a thorough job of addressing my concerns and suggestions, and the revised manuscript is much stronger and clearer for readers. Specifically, bias reduction is now addressed in the Methods section. The first paragraph in the Results section is much clearer now, with percentages based on the number of responses used in analyses. The revised Fig. 1 is much easier to interpret now with percentages added and additional caption text. CLMM results have been corrected, and parameter estimates are now explained in the main text. The revised Fig. 3 is also easier to read, and the caption now includes all pertinent information. The revised Discussion section is more concise, flows much better, and is more effective than the original.

I only have one suggestion & a few corrections, listed below:

Ln 93–96: I suggest changing this back to two separate sentences, as combining them results in a long sentence that’s tough to follow.

Ln 152: Correct the misplaced en dash (should be between “bird” and “window”).

Ln 166: Insert missing quotations at end of sentence.

Ln 410, 422, 423, 427, 493: Remove extraneous hyphens from the phrase “bird—window collisions”. (In these lines, both an en dash and hyphen are used, e.g. bird—-window collisions.)

Ln 422: It appears that the word “and” is missing between “houses” and “low-rise buildings”.

Ln 442: Delete extraneous period after “Fig 4”.

**Do you want your identity to be public for this peer review?** For information about this choice, including consent withdrawal, please see our Privacy Policy

Reviewer #2: No

---

## [Editor Report · Acceptance letter]

PONE-D-25-29481R1

PLOS One

Dear Dr. Lysyk,

I'm pleased to inform you that your manuscript has been deemed suitable for publication in PLOS One. Congratulations! Your manuscript is now being handed over to our production team.

Kind regards,

on behalf of

Dr. Teddy Lazebnik

Academic Editor

PLOS One